# Local synaptic inputs support opposing, network-specific odor representations in a widely projecting modulatory neuron

Xiaonan Zhang[1], Kaylynn Coates[2], Andrew Dacks[2], Cengiz Günay[3], J Scott Lauritzen[4], Feng Li[4], Steven A Calle-Schuler[4], Davi Bock[4,5], Quentin Gaudry[1]*

[1]Department of Biology, University of Maryland, College Park, United States; [2]Department of Biology, West Virginia University, Morgantown, United States; [3]School of Science and Technology, Georgia Gwinnett College, Lawrenceville, United States; [4]Janelia Farm Research Campus, Howard Hughes Medical Institute, Ashburn, United States; [5]Department of Neurological Sciences, Larner College of Medicine, University of Vermont, Burlington, United States

**Abstract** Serotonin plays different roles across networks within the same sensory modality. Previously, we used whole-cell electrophysiology in *Drosophila* to show that serotonergic neurons innervating the first olfactory relay are inhibited by odorants (Zhang and Gaudry, 2016). Here we show that network-spanning serotonergic neurons segregate information about stimulus features, odor intensity and identity, by using opposing coding schemes in different olfactory neuropil. A pair of serotonergic neurons (the CSDns) innervate the antennal lobe and lateral horn, which are first and second order neuropils. CSDn processes in the antennal lobe are inhibited by odors in an identity independent manner. In the lateral horn, CSDn processes are excited in an odor identity dependent manner. Using functional imaging, modeling, and EM reconstruction, we demonstrate that antennal lobe derived inhibition arises from local GABAergic inputs and acts as a means of gain control on branch-specific inputs that the CSDns receive within the lateral horn.
DOI: https://doi.org/10.7554/eLife.46839.001

*For correspondence:
qgaudry@umd.edu

Competing interests: The authors declare that no competing interests exist.

## Introduction

Virtually all neuronal circuits are subject to neuromodulation from both neurons intrinsic to a network and extrinsic centrifugal sources (*Katz, 1995*; *Lizbinski and Dacks, 2017*). In vertebrates, extrinsic modulation is often supplied by nuclei located deep within the brainstem that release a variety of transmitters such as norepinephrine (NE) (*Schwarz and Luo, 2015*), serotonin (5-HT) (*Charnay and Léger, 2010*; *Hornung, 2003*), dopamine (DA) (*Ikemoto, 2007*; *Lammel et al., 2008*), or acetylcholine (Ach) (*Wenk, 1997*). For example, the olfactory bulb (OB) in mammals receives a tremendous amount of centrifugal innervation (*Padmanabhan et al., 2018*) that can be critical for proper olfactory behavior (*Nunez-Parra et al., 2013*). However, by spanning and innervating most cortical and subcortical regions, modulatory systems target multiple points along the sensory-motor axis of functional circuits. A prominent view of such modulatory systems is that they provide a mechanism for top-down regulation of sensory processing (*Jacob and Nienborg, 2018*; *Thiele and Bellgrove, 2018*) and help coordinate activity across brain regions (*Melzer et al., 2012*). Modulatory systems are traditionally regarded as integrate-and-fire models where the neurons integrate synaptic inputs in their dendrites within their local nuclei and use action potentials to broadcast this signal to release sites across sensory networks. Such models imply that modulator release will be inherently correlated across distal targets. However, as virtually all axons are subject to pre-synaptic regulation

(*Miller, 1998*), it is likely that most centrifugal modulatory neurons are subject to local influences by the circuits that they infiltrate. This suggests that the local release of transmitters from such systems may instead be decorrelated across brain regions. Decorrelating transmitter release across brain regions is advantageous, as it would provide greater flexibility in how neuromodulation may be employed. How transmitter release is locally regulated in modulatory neurons and how signals propagate through their processes has been exceedingly difficult to study in vertebrate systems due to many contributing factors. First, the vertebrate cerebrum is large and imaging the extensive processes of such neurons across brain areas requires specialized tools (*Lecoq et al., 2014*; *Sofroniew et al., 2016*; *Stirman et al., 2016*; *Terada et al., 2018*). Second, individual modulatory neurons within the same brainstem nucleus are highly heterogeneous in their projection patterns (*Gagnon and Parent, 2014*; *van der Kooy and Kuypers, 1979*) making it difficult to assign activity across brain regions to individual cells. Additionally, the spatial pattern of extrinsic input activation can also be stimulus specific. For example, different odors can activate unique presynaptic terminals from piriform neurons that feed back into the OB (*Otazu et al., 2015*). But whether such odor responses in centrifugal inputs arise from the recruitment of different individual cortical neurons or from local axo-axonic interactions within the OB remains unknown. The spatial pattern of cholinergic input to the OB is also odor-specific and arises through similarly undescribed mechanisms (*Rothermel et al., 2014*).

The *Drosophila* brain is an ideal preparation to study how signals propagate through wide-field modulatory neurons because such cells are often stereotyped and can be genetically targeted across individual flies. For example, only one serotonergic neuron per hemisphere, termed the contralaterally-projecting serotonin-immunoreactive deuterocerebral neuron (CSDn), innervates both the first and second order olfactory neuropils in the fly brain (*Coates et al., 2017*; *Dacks et al., 2006*; *Roy et al., 2007*; *Sun et al., 1993*) (*Figure 1A*). CSDns can be targeted genetically (*Roy et al., 2007*) and affect various olfactory behaviors involving appetitive (*Xu et al., 2016*) and pheromonal odorants, including cis-vaccenyl acetate (cVA) (*Singh et al., 2013*). The modulation of cVA-evoked behavioral responses is especially interesting because the CSDns are thought to participate in top-down modulation and to have their effects mainly in the first olfactory relay, the antennal lobe (AL) (*Hill et al., 2002*; *Singh et al., 2013*; *Sun et al., 1993*). However, CSDn processes avoid the cVA-sensitive DA1 glomerulus (*Coates et al., 2017*; *Zhang and Gaudry, 2016*) and DA1 projection neuron (PN) odor responses are not modulated with strong CSDn stimulation (*Zhang and Gaudry, 2016*). Finally, whole-cell recordings show strong inhibition of the CSDn during stimulation with cVA (*Zhang and Gaudry, 2016*). This suggests that the modulatory effects of the CSDns on cVA-guided behavior may not occur in the AL, but rather in one of the other olfactory neuropil that the CSDns innervate. Because the CSDns express pre- and postsynaptic markers throughout their arborizations (*Zhang and Gaudry, 2016*), it is possible that transmitter release is locally regulated via inputs from their target networks (*Gaudry, 2018*) and that olfactory-mediated modulation occurs predominantly downstream of the AL. In this study we employed 2-photon calcium imaging, electron microscopy, and compartment modeling to show that the CSDns integrate synaptic inputs locally within their target regions giving rise to distinct odor evoked activity patterns within different olfactory neuropil.

## Results

To examine how CSDns contribute to olfactory processing across brain regions, we employed GCaMP6s (*Chen et al., 2013*) and 2-photon volumetric microscopy to characterize olfactory responses across their arbors. We initially imaged CSDn neurites in the AL (*Figure 1B*, *Figure 1—video 1*) and found that nearly all compounds in a diverse panel of odorants resulted in inhibition (*Figure 1C and D* and *Figure 1—figure supplement 1*). The only exception was ammonia, which produced a weak level of excitation as previously reported in whole-cell somatic recordings (*Zhang and Gaudry, 2016*). These results are consistent with previous studies demonstrating that odors generally inhibit the CSDns during electrophysiological somatic recordings (*Figure 1E*) (*Zhang and Gaudry, 2016*), which is due to prominent input from GABAergic local interneurons (LN) (*Coates et al., 2017*; *Zhang and Gaudry, 2016*).

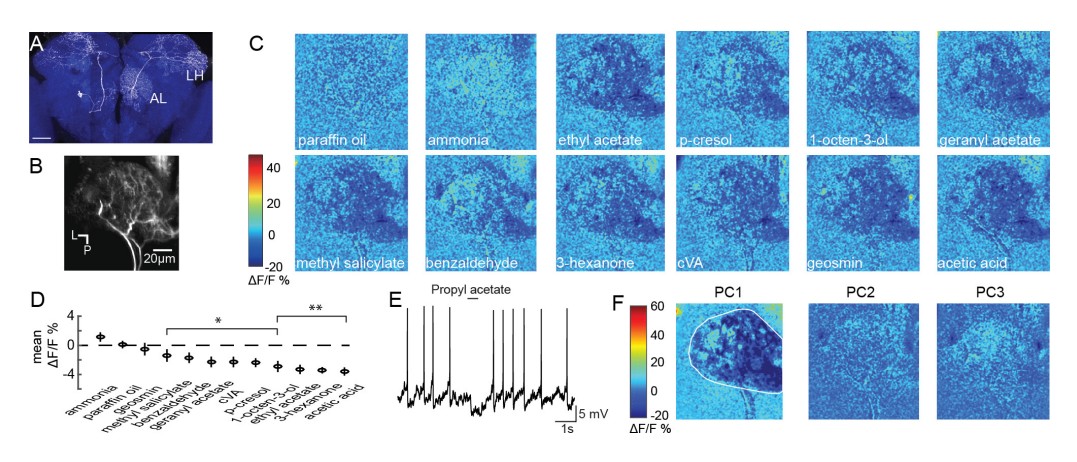

**Figure 1.** Olfactory stimulation inhibits CSDn processes in the AL. (**A**) A single CSDn expressing GFP shows processes in both the AL and the LH. White = GFP expression, blue = neuropil labeling via N-cadherin immunocytochemistry. Scale bar = 50 um. (**B**) Background fluorescence from GCaMP6s expression in CSDn neurites in the AL. L = lateral, p=posterior. The same notation is used in subsequent figures (**C**) Odor-evoked changes in calcium (ΔF/F) levels in the AL processes of the CSDn. Cooler colors (blue) represent decreases in calcium levels and warmer (red) colors show increases in calcium. Color bar values are set to the maximum and minimum pixel value across all odors. Images are generated from volumetric stacks by first averaging the z-stack for each volume during the trial to generate 2D frames, and then averaging across three frames during the peak of the odor response. Each odor was presented 3–4 times and final image is the mean frame of these trials. Odors are diluted $10^{-2}$ in paraffin oil (PA), which serves as a solvent control. Images are scaled and oriented as in *B*. (**D**) Odor responses ranked according to the strength of the observed inhibition (n = 10). From left to right, p=0.06, p=0.81, p=0.51, p=0.0097, p=0.0022, p=0.0052, p=0.0051, p=0.0012, p=0.0017, p=$1.82\times10^{-4}$, p=$2.95\times10^{-6}$, p=$2.50\times10^{-5}$. *Student's t-test.* *=p < 0.01, **=p < 0.001. (**E**) Odor-evoked inhibition observed in the CSDn soma via whole-cell patch-clamp recording. (**F**) Principal component analysis performed on the spatial pattern at the peak of the odor responses. White outline represents an ROI used to mask pixels outside of the AL for PCA. The first three PCs are shown. A structured response is only observed in PC1. Images are scaled and oriented as in *B*. The variance and SEM explained by PCs 1–3 are 60.3 ± 2.2%, 6.3 ± 0.4%, and 6.1 ± 0.3%, respectively.

DOI: https://doi.org/10.7554/eLife.46839.002

The following video and figure supplement are available for figure 1:

**Figure supplement 1.** CSDn calcium responses in the AL rescaled to emphasize excitation.

DOI: https://doi.org/10.7554/eLife.46839.003

**Figure 1—video 1.** Olfactory stimulation inhibits serotonergic CSDn neurites in the antennal lobe.

DOI: https://doi.org/10.7554/eLife.46839.004

Inhibition in the AL scales with increasing odor intensity and the spatial pattern of activation of GABAergic LNs is odor invariant (*Hong and Wilson, 2015*). Inhibition of the CSDn also scales with odor intensity (*Zhang and Gaudry, 2016*), but it is unknown if unique odors can recruit distinct spatial patterns of CSDn activity within its dendrites in the AL. We used principal component analysis (PCA) to examine the spatial profile of CSDn inhibition in response to our odor panel. This analysis revealed that only the first PC generated a structured image showing inhibition in the AL while PC2 captured the excitation resulting from stimulation with ammonia (*Figure 1F*). Thus, aside from ammonia, CSDn processes in the AL receive odor-invariant inhibition, and this inhibition scales with odor intensity (*Zhang and Gaudry, 2016*).

We next examined CSDn processes in the lateral horn (LH; *Figure 2A*), a brain region that receives direct input from the AL and mediates innate olfactory behaviors (*de Belle and Heisenberg, 1994*). Surprisingly, we found that every odorant in our panel produced excitation in the CSDn LH arbors (*Figure 2B*, *Figure 2—video 1*). Furthermore, PCA revealed that CSDn odor responses in the LH varied spatially (*Figure 2C*, *Figure 2—figure supplement 1*) and displayed a greater coefficient of variation compared to responses in the AL (*Figure 2D*). These results show that the processes of the CSDn have opposing responses to odor stimulation across different olfactory regions (*Figure 2E*). The CSDn has both pre- and postsynaptic sites in the LH (*Zhang and Gaudry, 2016*), so GCaMP signaling could represent either the activation of CSDn postsynaptic receptors or calcium influx at presynaptic release sites. Some GCaMP signaling in the LH must represent local activation of postsynaptic receptors in the CSDn since its AL processes are simultaneously inhibited. To assess

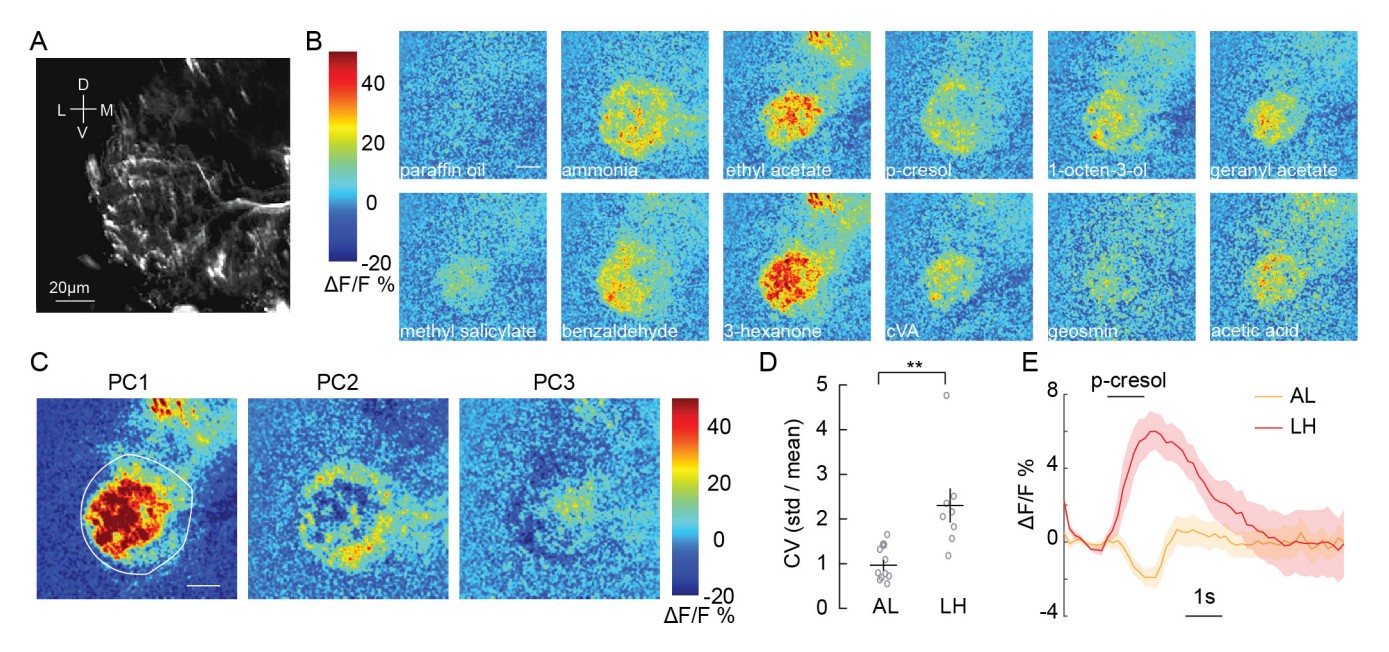

**Figure 2.** Olfactory stimulation excites CSDn processes in the LH. (A) Background fluorescence from GCaMP6s expression in CSDn neurites in the LH. (B) Odor-evoked changes in calcium observed in CSDn processes in the LH. Odor scale bar = 20 μm (C) Principal component analysis performed on the spatial pattern at the peak of the odor responses for a sample preparation. White outline represents an ROI used for masking during PCA.The first three PCs are shown. A structured response is observed in each PC. Images are scaled as in B. The variance and SEM explained by PCs 1–3 are 45.2 ± 2.6%, 12.4 ± 0.9%, and 8.6 ± 0.6%. (D) The coefficient of variation for odor responses in the AL and LH. n = 10, AL and n = 8, LH. p=0.0021, Student's t-test. The CV was calculated for each pixel in the odor response images and averaged across pixels for each preparation. (E) A comparison of the time series of ΔF/F responses in the AL and LH to a single odorant, p-cresol. The response is the averaged across 10 animals in the AL and eight in the LH. Error bars represent SEM.

DOI: https://doi.org/10.7554/eLife.46839.005

The following video and figure supplements are available for figure 2:

**Figure supplement 1.** Activation patterns in the LH are similar across preparations.
DOI: https://doi.org/10.7554/eLife.46839.006

**Figure supplement 2.** Olfactory stimulation activates presynaptic release sites in CSDn terminals in the LH.
DOI: https://doi.org/10.7554/eLife.46839.007

**Figure 2—video 1.** Olfactory stimulation excites serotonergic CSDn neurites in the lateral horn.
DOI: https://doi.org/10.7554/eLife.46839.008

whether this activity also correlates with synaptic release from the CSDn, we employed sytGCaMP6s, a variant of the calcium sensor that is tethered to synaptotagmin and trafficked to presynaptic release sites (*Cohn et al., 2015*). Olfactory stimulation showed increased sytGCaMP6s signaling (*Figure 2—figure supplement 2*), suggesting odorants likely evoke transmitter release in the LH processes of the CSDn.

To determine if spatial patterns of CSDn odor-evoked activity in the LH were odor-specific, we quantified the similarity between CSDn odor responses by first computing the spatial correlation between them (*Figure 3A*). We then calculated the Euclidean linkage distance between all odor correlations to illustrate which odors are most similarly encoded in the neurites of the CSDn in the LH (*Figure 3B*). How might odor specific responses arise in the CSDn processes in the LH? PNs are a potential source of excitatory input to the CSDns in the LH as they are cholinergic, and their axons segregate anatomically (*Jefferis et al., 2007*; *Tanaka et al., 2004*) and functionally (*Min et al., 2013*; *Seki et al., 2017*; *Strutz et al., 2014*) in this region. We compared the odor representations of PN and CSDn processes within this structure to determine if PNs may provide excitatory drive to the CSDn branches in the LH. The linkage distance between the spatial pattern of odor responses was highly correlated between PN and CSDn responses in the LH (*Figure 3C* and *Figure 3—figure*

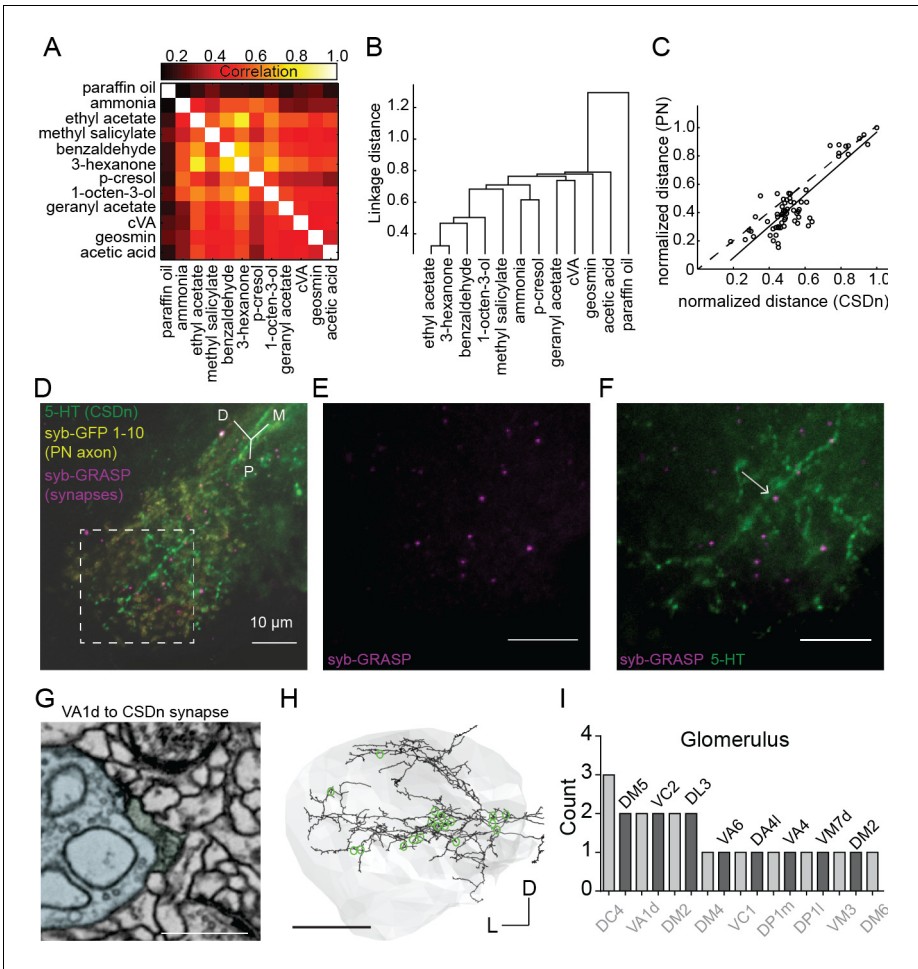

**Figure 3.** Projection neurons shape CSDn responses in the LH via direct synaptic input. (**A**) A cross correlation of the spatial profile of odor responses in CSDn LH processes. (**B**) Clustering analysis using the Euclidean distance between correlations for odor pairs from (**A**). (**C**) Regression analysis on correlation distances between odor responses in processes of the CSDn and PN axons in the LH ($R^2$ = 0.77, p=4.07*10$^{-22}$, n = 10 preparations for CSDn and n = 5 for PN responses). Distances were normalized to the max distance observed for CSDns and PNs. The 66 open circles represent the distances for each odor pair (12 odors and 66 pairwise comparisons). PN odor responses where measured using GCaMP6s and the GH146-Gal4 promoter. (**D–F**) GRASP images showing direct synaptic input from PNs onto CSDn terminals. Green = 5 HT antibody to label CSDn processes, yellow = syb GFP subunits 1–10 expressed in PN axons, magenta = syb GRASP labeling of synapses. Scale bars = 20 µm. (**G**) EM image of a direct VA1d PN synapse onto a CSDn neurite in the AL Scale bar = 500 nm. (**H**) EM reconstruction of the CSDn (black) arbors in the LH (grey boundary). Location of individual PN synapses onto the CSDn are marked in green. Scale bar = 25 um. (**I**) A total count of PN synapses onto the CSDn in the LH separated by glomerular identity. Glomeruli are listed above and below bars for clarity. DM2 is represented twice as two synapses arose from one DM2 PN and another synapse was identified from another DM2 PN. Counts taken from one female brain.

DOI: https://doi.org/10.7554/eLife.46839.009

The following figure supplement is available for figure 3:

**Figure supplement 1.** The axons of projection neurons in the LH have odor specific patterns of activation.

DOI: https://doi.org/10.7554/eLife.46839.010

*supplement 1*). These data suggest that PNs may provide direct synaptic input onto the CSDn locally in the LH. Using synaptobrevin GFP Reconstitution Across Synaptic Partners (syb:GRASP) (*Feinberg et al., 2008*; *Macpherson et al., 2015*) we observed a positive signal for PNs synapsing onto the CSDns (*Figure 3D–F*) in the LH. We further verified this synaptic connection by looking at the connectivity of a CSDn reconstructed within a whole fly brain EM dataset (*Zheng et al., 2018*).

PNs from at least 17 glomeruli (presynaptic PN tracings provided as a personal communication from Greg Jefferis, Phillipp Schlegel, Alex Bates, Marta Costa, Fiona Love and Ruari Roberts) show direct input to the CSDn throughout the LH (*Figure 3G–I*). Together these results suggest that the CSDns receive input from different cell classes in the AL and LH to shape its olfactory responses locally. In the AL, GABAergic LNs constitute a major input to the CSDns and drive inhibitory responses independent of odor identify. PNs make relatively few synaptic connections with the CSDns in the AL (*Coates et al., 2017*). In the LH, olfactory responses in the CSDn neurites are excitatory, odor specific, and are likely in part shaped by direct PN input.

The function of a neuron is often dictated by the manner in which synaptic inputs are integrated across its dendritic arbor. We therefore asked whether the AL and LH neurites of the CSDn function as electrotonically independent compartments within the same cell, or if signals propagate between regions during odor sampling. To examine how voltage spreads throughout the CSDn, we built a passive compartmental model based on an anatomical reconstruction of the CSDn generated from the whole fly brain EM dataset (*Carnevale and Hines, 2006*; *Zheng et al., 2018*) (*Figure 4A* and *Figure 4—figure supplement 1*). While such models ignore the contributions of active conductances or variations in passive properties across dendrites, they can nevertheless provide insight into signal propagation and help formulate hypotheses. The model was generated by adjusting the membrane capacitance ($C_m$), the membrane conductance ($g_{leak}$), and the axial resistance ($R_a$) so that simulated current injections into the model soma matched physiological responses taken in vivo (*Figure 4B*). A wide range of models with varying properties provided reasonable fits to the in vivo CSDn recordings. In these models, injecting simulated hyperpolarizing IPSPs into the AL resulted in varying amounts of spread throughout the neurites of the CSDn. In some models, the hyperpolarization was constrained only to the local injection site (*Figure 4C*), while in other models the hyperpolarization spread throughout the AL (*Figure 4D*). However, whole-cell recordings in vivo show that the inhibition arising from the AL indeed spreads to the soma (*Zhang and Gaudry, 2016*). Several of the models that we generated displayed this property. Importantly, all models that displayed somatic inhibition when current was injected into the CSDn processes in the AL also showed inhibition in the LH (*Figure 4E*). These results suggest that the geometry and passive properties of the CSDn allow the propagation of inhibition from the AL to the LH during olfactory stimulation.

The propagation of voltages through the dendrites of neurons is not always symmetrical and can be biased by impedance mismatches as well as differences in the diameters of intersecting branches (*Stuart et al., 2016*). We therefore asked if voltage changes generated in the LH would propagate as efficiently to the AL compared to the opposite direction. We injected current into the either the LH or the AL of the model to elicit voltage changes of similar magnitudes and calculated the proportion of the signal that propagated to the other region. This approach revealed that the geometry of the CSDn arbors favor the propagation of voltage signals from the AL preferentially to the LH (*Figure 4F*).

If voltage signals preferentially propagate from the AL to the LH, one would predict that isolating the CSDn processes in the LH from the rest of the CSDn would boost any odor evoked excitation due to local input within the LH. We therefore used 2-photon laser ablation to sever the CSDn neurites just proximal to their entrance into the LH. This manipulation isolates the LH processes of the CSDn from the AL, while still allowing these neurites to respond to synaptic input within the LH (*Figure 4G and H*). Olfactory responses to all odors tested increased in the processes of the CSDn in the LH following the removal of AL inhibition (*Figure 4I,J and K*). As a control for the non-specific damage of laser ablation, we severed the branches of the CSDn in the contralateral hemisphere and found that this had no impact on odor responses in the intact branches of the CSDn in the ipsilateral LH (*Figure 4K*). These results demonstrate that during normal odor sampling, inhibition from the AL propagates to suppress olfactory responses in the LH.

## Discussion

There has been a recent interest in characterizing the structures that provide input to serotonergic neurons in vertebrates in an attempt to understand the types of processes that might impact 5-HT release (*Sparta and Stuber, 2014*). Such studies relied on genetically restricted retrograde labeling to identify regions that provide monosynaptic inputs to serotonergic raphe neurons (*Ogawa et al., 2014*; *Pollak Dorocic et al., 2014*; *Weissbourd et al., 2014*). Our functional approach of calcium

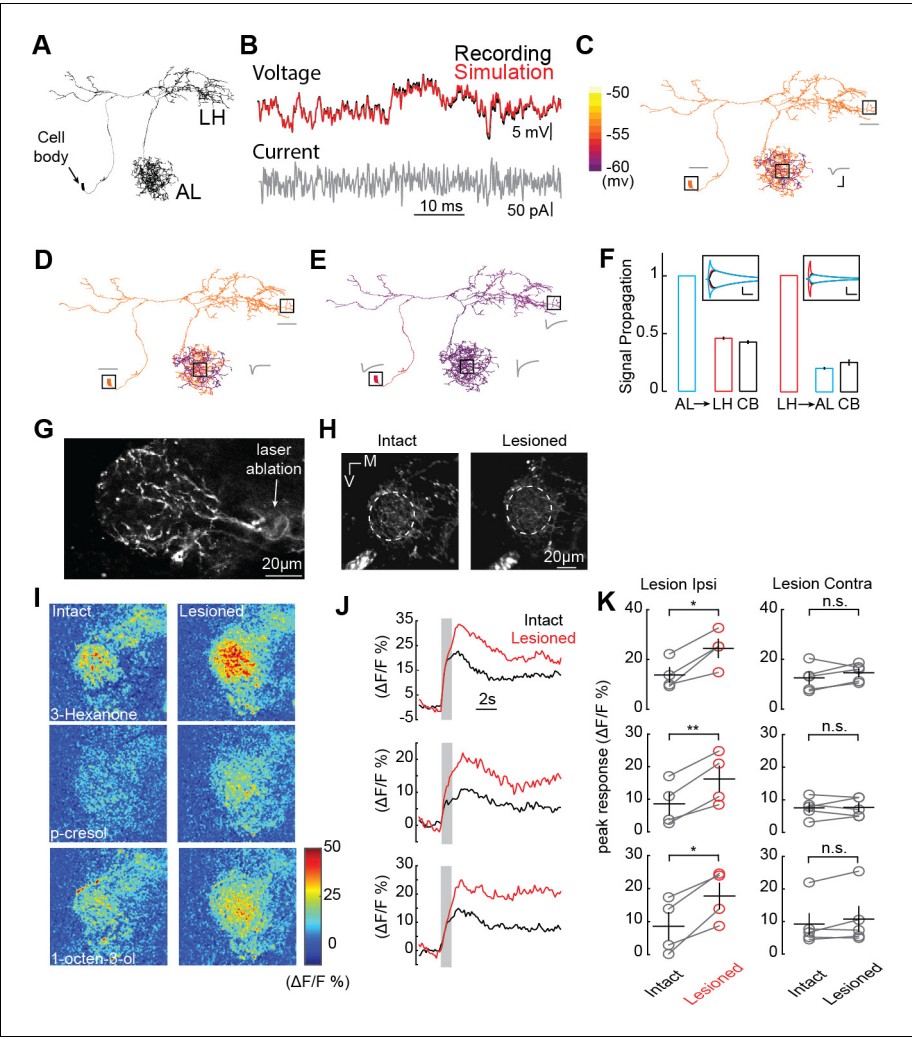

**Figure 4.** AL inhibition suppresses CSDn responses in the LH. (**A**) Morphological EM reconstruction of the CSDn arborizations used for a biophysical model consisting of 6056 compartments. (**B**) Somatic membrane potential changes in response to a white-noise current injection during an in vivo CSDn recording compared to responses of a passive compartmental model, to the same stimulus, fitted by optimizing three anatomical parameters, $C_m$ ($\mu F/cm^2$), $g_{leak}$ ($S/cm^2$), and $R_a$ ($\Omega cm$), (**C**) An example of a model with well-fit somatic responses where a hyperpolarizing current injection in the AL resulted in highly localized voltage changes in AL only. A time series of voltage responses are shown for the soma, AL, and LH as gray traces near those structures. $C_m = 4.8$, $g_{leak} = 3094.3$, and $R_a = 149.0183$. Horizontal scale bar = 5 ms and vertical scale bar = 5 mV. (**D**) A similar model where injection of the current in the AL causes a voltage change throughout a greater portion of the AL. $C_m = 4.8$, $g_{leak} = 2911.3$, and $R_a = 28.83$. Scale bars as in *C*. (**E**) A sample model where current injection into the AL causes both somatic and LH voltage changes. $C_m = 1.8$,$_w g_{leak} = 77.33$, and $R_a = 0.12$. See **Table 1** for full model properties .Scale bars as in *C*. (**F**) The proportion of voltage signal observed in the LH, AL, and cell body (CB) when voltage changes are induced in each region. The same model as in panel *E* was used to generate the data. *Left*, voltage steps are induced into the AL and voltage deflections are reported in the LH and CB. *Right*, voltage steps are induced in the LH and their effects are measured in the AL and CB. Horizontal scale bar = 10 ms and vertical scale bar = 10 mV. (**G**) A 2P image of basal GCaMP6s signal showing the effects of laser ablation in the CSDn LH neurite. (**H**) Basal GCaMP6s signals remain in the LH after laser ablation. (**I**) ΔF/F responses of CSDn neurites in the LH to a set of odorants before and after laser ablation. (**J**) A times series of the changes in calcium levels in response to odorants in (**I**). (**K**) *Left*, quantification of odor response amplitudes in CSDn LH neurites before and after laser ablation. *Right*, control responses when the contralateral processes of the CSDn where ablated. *Left column, top to bottom*, p=0.0187, p=0.0067, p=0.0154; *Right column, top to bottom* p=0.2298, p=0.8958, p=0.1611. *=p < 0.05, **=p < 0.01, n.s. = not significant and p>0.05, paired Student's t-tests.
DOI: https://doi.org/10.7554/eLife.46839.011

*Figure 4 continued on next page*

*Figure 4 continued*

The following figure supplements are available for figure 4:

**Figure supplement 1.** Sampling strategy for measuring CSDn axon radius.

DOI: https://doi.org/10.7554/eLife.46839.012

**Figure supplement 2.** cVA excites CSDn processes in the lateral horn.

DOI: https://doi.org/10.7554/eLife.46839.013

imaging across brain regions is complementary but has the advantage of revealing where signals are integrated and determining whether the net effect of that integration within is excitatory or inhibitory. Here, we show that a serotonergic neuron with broad arbors integrates locally at multiple neuropil along to the olfactory pathway in *Drosophila*. Interestingly, synaptic inputs at the first and second order processing stages of olfaction impose opposite polarity responses in this neuron and likely decorrelate synaptic release in distinct target regions. Inhibition dominates responses in the AL and odor-specific excitation is prominent in the downstream LH. This suggests that across a single widely projecting modulatory neuron, branches within distinct neuropils can operate in different manners.

## Local regulation of release in modulatory neurons

Classical methods of studying the activation of modulatory systems during sensory processing in vertebrates rely heavily on recording extracellular spikes in the nuclei that house the neurons' somas. However, invertebrate studies involving motor systems have shown compartmentalization of neurons spanning distant neuropils (*Nikitin et al., 2013*; *Sasaki et al., 2007*). Our results demonstrate that the activity within a single serotonergic neuron can vary across neuropils involved in processing the same sensory modality. As a given modulator may perform distinct functions in different brain regions, local regulation of release allows these functions to be employed independently. For instance, 5-HT in the OB indirectly inhibits OSN terminals and has been proposed to serve a gain control function (*Petzold et al., 2009*), while in the piriform cortex 5-HT has no effect on stimulus input, but rather only decreases spontaneous activity (*Lottem et al., 2016*). Additionally, 5-HT has different effects on mitral cells in the main versus mitral cells in the accessory olfactory bulb (*Huang et al., 2017*). Local regulation of 5-HT release would allow these processes to be engaged in an independent and combinatorial manner, and thus allow for a greater net overall modulatory capacity.

Local regulation and decoupling of modulator release across synaptic sites is not unique to invertebrates and has been implicated in the normal function of the mammalian DA system. First, DA release is only partially correlated with firing activity, and release can be locally evoked in the absence of spiking in DA cells (*Floresco et al., 1998*). Additionally, local inactivation of the nucleus accumbens (NA) decreases DA release in the NA without impacting DA neuron firing (*Jones et al., 2010*). This has led to the theory that DA release can signal both motivation and reward prediction errors (RPE) on similar timescales (*Hamid et al., 2016*). Dopamine release related to motivation is thought to be shaped by local presynaptic mechanisms while dopamine related to RPE correlates more strongly with the firing properties of DA neurons (*Berke, 2018*). Measuring serotonergic transmission across release sites is more difficult compared to dopamine (*Dankoski and Wightman, 2013*), nevertheless, it is highly likely that 5-HT is also regulated by local presynaptic mechanisms (*Egashira et al., 2002*; *Schlicker et al., 1984*; *Tao and Auerbach, 1995*; *Threlfell et al., 2004*). Whether there is local regulation of 5-HT release in the vertebrate olfactory system is more controversial. EM reconstructions of raphe terminals in the OB have failed to reveal postsynaptic densities in raphe axons (*Gracia-Llanes et al., 2010*; *Suzuki et al., 2015*). This may be because such input is extra synaptic (*Gaudry, 2018*), as is the case with GABA$_B$ receptors in the raphe nucleus proper (*Varga et al., 2002*).

## Multi-dendritic processing

Local integration in the AL and LH allow the CSDn to independently process and shape sensory information at multiple points in the early olfactory pathway of the fly. Specifically, we found that PN axons directly excite CSDn terminals in the LH in an odor specific manner while CSDn branches in

the AL are inhibited. Previous studies have shown that the CSDn soma is also inhibited by odors and that stimulation of the CSDn has little impact on olfactory circuitry in the AL (*Zhang and Gaudry, 2016*), despite the CSDn being critical for normal olfactory behavior (*Singh et al., 2013*; *Xu et al., 2016*). Our current study resolves this issue by suggesting instead that the CSDn modulates behavior by affecting odor-processing in the LH (*Figure 4* and *Figure 4—figure supplement 2*), which has previously been unexplored. What then is the purpose of odor-evoked inhibition in the AL? Inhibition of CSDn processes in the AL does not appear to depend on odor identity as shown in this study, but rather correlates with the total amount of ORN activity that an odor elicits (*Zhang and Gaudry, 2016*). This can vary with both odor identity or concentration. Our compartmental modeling shows that AL and LH processes of the CSDn are electrotonically connected, but that voltage preferentially passes from the AL to the LH. As CSDn inhibition scales with increasing odor strength due to robust GABAergic LN input in the AL (*Coates et al., 2017*; *Zhang and Gaudry, 2016*), this inhibition shunts LH responses proportionally. This configuration ultimately allows olfactory modulation to be odor specific in the LH while being less dependent on odor concentration. Multi-dendritic computing is critical for processing other sensory modalities as well (*Ranganathan et al., 2018*), but most notably vision (*Jones and Gabbiani, 2010*; *Koren et al., 2017*). Synaptic integration between dendrites is used to compute object motion across the visual field on a collision course with the observer. Interestingly, the dendrites of starburst amacrine cells in the mammalian retina express mGluR to isolate dendritic compartments thus preventing integration to non-preferred stimuli while enabling regulated integration specifically to preferred directions of motion (*Koren et al., 2017*). Our laser ablation experiments demonstrate that CSDn neurites influence one another during olfaction, but it is intriguing that the magnitude of such coupling could be state dependent and regulated by active conductances.

## Top-down versus bottom-up neuromodulation

The CSDn was originally proposed to participate in top-down modulation by transmitting higher order sensory information from the LH to the AL (*Hill et al., 2002*; *Sun et al., 1993*). However, direct evidence for top-down modulation via the CSDn has never been demonstrated. Our results suggest that during olfaction, the CSDn acts more in a bottom-up fashion where responses in the AL have a greater impact on downstream processing in the LH. Transmitter release during olfactory sampling putatively occurs only later in the sensory processing stream rather than at the earliest stages. Interestingly, the CSDn has numerous release sites in the AL (*Coates et al., 2017*; *Sun et al., 1993*; *Zhang and Gaudry, 2016*) and CSDn derived 5-HT directly inhibits several classes of neurons in the AL (*Zhang and Gaudry, 2016*). But how are CSDn release sites activated in the AL? Olfaction clearly inhibits both the processes of the CSDn in the AL as well at its spike initiation site (*Zhang and Gaudry, 2016*). However, it is likely that the CSDn also receives input from unidentified non-olfactory sources that excite the neuron's spike initiation site allowing it to modulate in a top-down manner. Further reconstruction of the CSDns inputs in the EM data set will reveal candidates for further physiological evaluation. Thus, non-olfactory stimulation of the CSDn may be consistent with top-down modulation and would constitute one mechanism by which a broadly arborizing modulatory neuron may be multifunctional depending on the source of its excitatory drive. Multifunctional neurons have been well described in central pattern generating networks (*Briggman and Kristan, 2008*) but are only recently becoming appreciated with regards to neuromodulation (*Berke, 2018*). Local synaptic interactions within the AL could shape 5-HT release as well to alter olfactory coding in a glomerulus-specific fashion (*Kloppenburg and Mercer, 2008*). Our findings in *Drosophila* suggest that integration into local circuits allows modulatory cells greater flexibility in how they participate in sensory processing and may be a feature that is often overlooked when assessing the function of these critical components of the central nervous system.

## Materials and methods

### Odors and odor delivery

Odors were presented as previously described (*Zhang and Gaudry, 2016*). In brief, a carrier stream of carbon-filtered house air was presented at 2.2 L/min to the fly continuously. A solenoid was used to redirect 200 ml/min of this air stream into an odor vial before rejoining the carrier stream, thus

diluting the odor a further 10-fold prior to reaching the animal. All odors were diluted 1:100 in paraffin oil (J.T. Baker VWR #JTS894), except for acids, which were diluted in distilled water. All odors were obtained from Sigma Aldrich (Saint Louis, MO) except for cVA, which was obtained from Pherobank (Wageningen, Netherlands). cVA was delivered as a pure odorant. In our olfactometer design, the odor vial path was split to 16 channels each with a different odor or solvent control. Pinch valves (Clark Solutions, Hudson MA part number PS1615W24V) were used to select stimuli between each trial. Each odor was presented sequentially one trial at a time. Each odor was presented 3–4 times within a preparation and the mean of these responses were then averaged across animals.

| Odors | Supplier |
| --- | --- |
| Paraffin oil | J.T.Baker CAS: 8012-95-1 |
| Ammonium hydroxide | Sigma-Aldrich CAS: 1336-21-6 |
| Ethyl acetate | Sigma-Aldrich CAS: 141-78-6 |
| Methyl salicylate | Sigma-Aldrich CAS: 119-36-8 |
| Benzaldehyde | Sigma-Aldrich CAS: 100-52-7 |
| 3-hexanone | Sigma-Aldrich CAS: 589-38-8 |
| p-cresol | Sigma-Aldrich CAS: 106-44-5 |
| 1-octen-3-ol | Sigma-Aldrich CAS: 3391-86-4 |
| Geranyl acetate | Sigma-Aldrich CAS: 105-87-3 |
| cVA | Pherobank, Wijk bij Duurstede, Netherlands |
| geosmin | Sigma-Aldrich CAS: 16423-19-1 |
| acetic acid | Sigma-Aldrich CAS: 64-19-7 |

## Fly Genotypes

The following *Drosophila* genotypes were used in this study: w; UAS-GCaMP6s; R60F02-Gal4, UAS-GCaMP6s, w; NP2242-Gal4/Cyo; R60F02-Gal4, UAS-GCaMP6s, w; GH146-Gal4/+; UAS-GCaMP6s/+and w; UAS-CD4:spGFP11/Q-GH146; QUAS-syb:spCFP1-10/R60F02-Gal4. PN odor responses were measured in flies expressing the GH146-Gal4 and UAS-GCaMP6s transgenes. To produce the singleton CSDn in *Figure 1A*, we used UAS-myrGFP, QUAS-mtdTomato-3xHA/+; trans-Tango/+; Gal4-MB465C/+which occasionally labeled individual CSDns. Flies were generated from the following stocks.

| Genotype | Source | RRID |
| --- | --- | --- |
| 20XUAS-IVS-GCaMP6s (attP40) | BDSC | BDSC_42746 |
| GMR60F02-GAL4 (attP2) | BDSC | BDSC_48228 |
| 20XUAS-IVS-GCaMP6s (VK00005) | BDSC | BDSC_42749 |
| GH146-Gal4 | BDSC | BDSC_30026 |
| UAS-CD4:spGFP11 | BDSC | BDSC_64315 |
| GH146-QF2 | BDSC | BDSC_66480 |
| QUAS-syb:spCFP1-10 | Marco Gallio, Northwestern University | |
| UAS-myrGFP, QUAS-mtdTomato-3xHA; trans-Tango | BDSC | BDSC_77124 |
| MB465C-Gal4 | BDSC | BDSC_68371 |
| NP2242-Gal4 | DGRC | 104134 |

BDSC, Bloomington Drosophila Stock Center, Bloomington Indiana. DGRC, Kyoto Stock Center, Kyoto Japan.

## Calcium imaging of odor-evoked activity

Female flies aged 3–5 weeks post-eclosion and reared at room temperature were used. In vivo imaging experiments were performed at room temperature. The brain was constantly perfused with

saline containing (in mM): 103 NaCl, 3 KCl, 5 N-tris(hydroxymethyl)methyl-2- aminoethane-sulfonic acid, eight trehalose, 10 glucose, 26 NaHCO3, 1 NaH2PO4, 2 CaCl2, and 4 MgCl2 (adjusted to 270– 275 mOsm). The saline was bubbled with 95% O2/5% CO2 and reached a pH of 7.3. 920 nm wavelength light was used to excite GCaMP6s under two-photon microscopy. The microscope and data acquisition were controlled by ThorImage 3.0 (Thorlabs, Inc). Three-dimensional volumes of neuropil were used to measure odor responses. Volumes consisted of 6–8 frames separated by 5–6 µm in the z plane between frames. The volumes were scanned at 60 frames/second and thus a 6–7 complete volumes were imaged each second. Single imaging trials consisted of 90 volumes at a resolution of 256 × 256 pixels. Odors were delivered for 1 s after the first 2 s of each trial. An 80 s interval between each trial was applied to allow replenishing of the odor vial head space and to prevent photobleaching of GCaMP. Odors were presented sequentially such that each odor was presented every 16 min.

To quantify odor responses, we first generated the mean z-projection for every volume within a trial. Calcium transients (ΔF/F) were then measured as changes in fluorescence, in which ΔF/F was calculated by normalizing the fluorescence brightness changes over the baseline period (the first 2 s of each trial before the odor delivery). To reduce noise, a Gaussian low-pass filter of 4 × 4 pixels in size was then applied to raw ΔF/F signals prior to further analysis.Some trials contained motion artifacts that were removed using StackReg plugin for ImageJ. This motion correction was done on an individual trial basis. The frame containing the peak response was identified by plotting the ΔF/F in an ROI as a function of frame number. The peak calcium signal for each trial was computed as the average of three consecutive frames centered on the frame of the peak response. We set the peak window the same for all trials within a preparation. For each odor stimulus, data were pooled by averaging the peak odor-evoked calcium signal across 3–4 repeats. All imaging data are available at https://zenodo.org/record/3347197#.XTq2PpNKjUl.

## Statistical analysis of calcium imaging

PCA was applied on the spatial pattern of the peak calcium signal (ΔF/F) in each trial (the mean frame of 2–3 volumes during the peak odor response). PCA was computed using the 'princomp' function in Matlab (Mathworks, Natick, MA). First an odor response was generated for each trial as described above as the mean response of 2–3 volumes during the peak response and averaged across 3–4 odor presentations per fly. The resulting image for each odor was converted into a 1-dimensional array consisting of 65,536 elements generated from the original 256 by 256 image). The odor response arrays were inserted as columns for the PCA input matrix. This matrix thus consisted of 12 columns, each representing an odor response. The output PCs were later reshaped back to 256 by 256 matrices for display purposes. The coefficient of variation was also calculated on the same vectorized odor response matrix so that a CV was generated for each pixel across odor responses and then averaged within that preparation. Thus, our reported CV incorporates variance for both the spatial profile and amplitude components of the responses. Correlations between odor responses within the LH were also performed using the same mean responses that served as inputs to the PCA. To calculate the diversity of response patterns to different odors we applied Linkage Hierarchical Clustering. The Euclidean distances between correlation pairs was calculated used as a parameter for clustering. Distances were normalized to the maximum distance within each preparation and then averaged across flies. A regression analysis was applied to compare the similarity between the odor response patterns of the CSDn and PNs in the LH. Our data set consisted of 12 odorants and thus 66 Euclidian distances were calculated for the regression. The p-value and the square of correlation coefficient ($R^2$) were calculated as the indicator of similarity for each odor response pattern pairs evoked. Two-tailed paired t-tests were performed for all comparisons between before laser ablation and after laser ablation within the same group. All statistical functions were applied in Matlab.

## CSDn EM reconstruction

The CSDn was identified and partially reconstructed in the female adult fly brain (FAFB) dataset as described in *Zheng et al. (2018)* using CATMAID (*Saalfeld et al., 2009*; *Schneider-Mizell et al., 2016*), available at https://catmaid-fafb.virtualflybrain.org. The CSDn reconstruction from the cell body along the primary and secondary arbors leading into the LH as well as tertiary and quaternary

branches into the lateral horn were reviewed by a second observer back to the primary branch as previously described (*Zheng et al., 2018*). For the multi-compartmental model, measurements of the CSDn branch radius (to inform axial resistance parameters) were taken at seven locations along the primary arbor between the contralateral AL and protocerebrum, five locations along the second order branch leading into the lateral horn and six locations along primary and secondary branches within the lateral horn. For each location, 20–50 measures of axon radius were taken from consecutive tissue sections. Data on the projection neurons that are pre-synaptic to CSDn in the FAFB dataset were provided as a personal communication from Greg Jefferis, Phillip Schlegel, Alex Bates, Marta Costa, Fiona Love and Ruari Roberts.

## Model construction

A multi-compartmental conductance-based computer model of the CSDn neuron was constructed by taking its electron micrograph reconstruction and importing it into the Neuron simulator (*Carnevale and Hines, 2006*). An initial reconstruction contained more than 60,000 nodes, but a simplified version of it was generated with 6056 compartments in the Neuron simulator that retained its basic anatomical and electrotonic structure. The passive cable parameters (axial resistance $R_a$, leak conductance $g_{pas}$, leak reversal $E_{pas}$, and specific capacitance $C_m$) were fitted using Neuron's RunFitter algorithm. For fitting, we used recorded responses to stimuli of current-clamp and voltage-clamp steps and current-clamp white noise generated by Matlab (Mathworks, Natick, MA). Whole-cell recordings of the CSDn were performed as previously described (*Zhang and Gaudry, 2016*). The series resistance for CSDn recordings was approximately 10 MΩ, and input resistance was 500–600 MΩ. The pipette resistance was between 8 and 10 MΩ. The reversal potential of the CSDn was −45 mV. While fitting, parameters were restricted by physiological ranges ($R_a$ between 0.0001–5000 Ωcm, $C_m$ between 0.1–2 μF/cm$^2$, $g_{leak}$ between $10^{-6}$-0.1 S/cm$^2$, and Epas had unlimited range in mV, *Supplementary file 1*). For a given model, each compartment had the same values for Ra, gpas, Epas, and Cm. Only the difference in process thickness changed the passive properties of the compartments. The resulting passive model of the CSDn neuron was simulated using Neuron's default integration method with a time step of 0.025 ms. The properties of the model used to generate the data in *Figure 4E and F* are shown in *Table 1*.

To investigate how the passive signal travels from the AL to the other parts of the CSDn, we randomly picked ten spots within the AL and injected a square wave of current into the model to elicit a maximum voltage response ranging from about 2 to 20 mV. This was repeated six times in each condition. We monitored the voltage changes in the windows shown in *Figure 4*. The windows we set were for the cell body, the center of AL (three randomly selected monitor sites) and the LH (three randomly selected monitoring sites). The average voltage changes of each window were shown in the *Figure 4*. To investigate how signals preferentially propagated along the CSDn, the same method was applied to LH, so that current was injected in the LH and voltage change windows of interests across the CSDn were kept the same. The process was then repeated with current injection into the AL. The CSDn dendritic architecture does not represent a unique solution to asymmetrical voltage propagation. We were able to reproduce a similar phenomenon using a simpler three compartment model consisting of only a soma, an AL branch and a LH branch. The only requirement is to have differences in the resistances of two adjoining dendrites as predicted by cable theory (*Rall and Agmon-Snir, 1998*).

**Table 1.** Morphological statistics of CSDn neuron sections used in *Figure 4E and F*.

| | L [μm] | Diam [μm] | Area [μm$^2$] | R$_i$ [Mς] | C$_m$ [pF] | g$_{pas}$ [nS] |
|---|---|---|---|---|---|---|
| Whole cell | 11031.21 | 0.05 | 1611.09 | 370.28 | 16.11 | 4.26 |
| Cell body | 308.15 | 0.27 | 258.89 | 0.31 | 2.59 | 0.68 |
| LH | 3758.09 | 0.04 | 474.12 | 169.05 | 4.74 | 1.25 |
| AL | 6245.41 | 0.04 | 787.66 | 281.13 | 7.88 | 2.08 |

L, total length of section branches; Diam., length-weighted equivalent diameter of section; Area, surface area, R$_i$, resistance of from beginning to middle of section; C$_m$, total maximal capacitance of section (does not consider decay of voltage); g$_{pas}$, total area-scaled leak conductance of the equivalent section.

DOI: https://doi.org/10.7554/eLife.46839.015

## Laser transection

The transection window was guided by the Gcamp6s basal fluorescence at 920 nm, at about 20 µm before the CSDn neurites enter the lateral horn. An 80 mW laser pulse, which consisted of 10 repetitions of continuous frame scanning with 8 µsec of pixel dwell time, at 800 nm was then applied onto this window. A total estimated energy of 0.05 J was thus applied. Successful transection usually resulted in a small cavitation bubble (shown in *Figure 4*).

## KCl induction of GRASP and immunohistochemistry

In brief, brains used for the induction of syb:GRASP were dissected and rinsed three times with a KCl solution (*Macpherson et al., 2015*). The brains were then fixed in 4% paraformaldehyde for 20 min. We used the following primary and secondary antibodies at the indicated dilutions: 1:1000 rabbit anti-5HT Sigma (S5545), 1:50 chicken anti-GFP Invitrogen (A10262), 1:100 mouse anti-GFP (referred as anti-GRASP, Sigma #G6539, ref:3), 1:500 rat anti-N-Cadherin (Developmental Studies Hybridoma Bank, DN-Ex #8), 1:250 Alexa Fluor 633 goat anti-rabbit (Invitrogen, A21071), 1:250 Alexa Fluor 488 goat anti-chicken (Life Technologies, A11039), 1:250 Alexa Fluor 568 goat anti-mouse IgG (Life Technologies, A11004) and 1:1000 Alexa Fluor 647 donkey anti-rat IgG (AbCam, ab150155). Brains were mounted and imaged in Vectashield mounting medium (Vector Labs). All steps were performed at room temperature. Confocal z-stacks for the syb:GRASP experiments were collected with a Zeiss LSM710 microscope using a 63 × oil immersion lens and the z-stack of GFP expression in *Figure 1A* was collected with an Olympus FV1000s using a 40x oil-immersion lens.

## Acknowledgements

The authors would like to thank Greg Jefferis, Phillip Schlegel, Alex Bates, Marta Costa, Fiona Love and Ruairi Roberts for providing access to tracings of PNs in the FAFB dataset, as well as Jay Milam for assistance with confocal scans and Tom Kazimiers, Andrew Champions, Chris Barnes and Albert Cardona for CATMAID support and Mert Erginkaya for assistance with tracing review. We would also like to thank Philipp Schlegel for developing the software Pymaid and the Python code to analyze the CSDn reconstruction and Kevin Daly for useful comments on the manuscript. This work was supported by a Whitehall Foundation Grant and an NIH R21 to QG, an NIH DC 016293 to AMD and QG, and Georgia Gwinnett College VPASA Seed Fund partially supported CG. The contributions of AMD and KEC in this project were supported in part by the Janelia Visiting Scientist Program.

## Additional information

### Funding

| Funder | Grant reference number | Author |
|---|---|---|
| Whitehall Foundation | | Quentin Gaudry |
| National Institute on Deafness and Other Communication Disorders | R21 DC015873-02 | Quentin Gaudry |
| National Institute on Deafness and Other Communication Disorders | RO1 DC016293 | Andrew Dacks Quentin Gaudry |
| Georgia Gwinnett College | VPASA Seed Fund | Cengiz Günay |
| Janelia Visiting Scientist Program | | Andrew Dacks Kaylynn Coates |

The funders had no role in study design, data collection and interpretation, or the decision to submit the work for publication.

### Author contributions

Xiaonan Zhang, Conceptualization, Formal analysis, Data collection; Kaylynn Coates, Conceptualization, Formal analysis, Writing—original draft, Writing—review and editing, Data

collection; Andrew Dacks, Supervision, Funding acquisition, Writing—original draft, Project administration, Writing—review and editing; Cengiz Günay, Formal analysis, Methodology, Writing—review and editing; J Scott Lauritzen, Feng Li, Methodology, Data collection; Steven A Calle-Schuler, Data collection; Davi Bock, Supervision, Methodology; Quentin Gaudry, Conceptualization, Supervision, Funding acquisition, Writing—original draft, Project administration, Writing—review and editing

### Author ORCIDs
Xiaonan Zhang  http://orcid.org/0000-0003-3181-3648
Kaylynn Coates  http://orcid.org/0000-0003-2592-8908
Andrew Dacks  https://orcid.org/0000-0002-6805-4211
Cengiz Günay  http://orcid.org/0000-0001-7586-571X
Feng Li  https://orcid.org/0000-0002-6658-9175
Davi Bock  http://orcid.org/0000-0002-8218-7926
Quentin Gaudry  https://orcid.org/0000-0002-6869-1253

### Decision letter and Author response
Decision letter https://doi.org/10.7554/eLife.46839.021
Author response https://doi.org/10.7554/eLife.46839.022

## Additional files

### Supplementary files
• Supplementary file 1. A table of the parameters and their values used to construct the passive compartmental model of the CSDn.
DOI: https://doi.org/10.7554/eLife.46839.017

• Transparent reporting form
DOI: https://doi.org/10.7554/eLife.46839.018

### Data availability
All data are publicly available at https://zenodo.org/record/3347197#.XTq2PpNKjUI.

The following previously published dataset was used:

| Author(s) | Year | Dataset title | Dataset URL | Database and Identifier |
|---|---|---|---|---|
| Zheng Z, Lauritzen JS, Perlman E, Robinson CG, Nichols M, Milkie D, Torrens O, Price J, Fisher CB, Sharifi N, Calle-Schuler SA, Kmecova L, Ali IJ, Karsh B, Trautman ET, Bogovic JA, Hanslovsky P, Jefferis GSXE, Kazhdan M, Khairy K, Saalfeld S, Fetter RD, Bock DD | 2018 | fafb | https://catmaid-fafb.virtualflybrain.org | CATMAID, fafb |

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
