## [Decision Letter]

Thank you for submitting your article "Local synaptic inputs support opposing, network-specific odor representations in a widely projecting modulatory neuron" for consideration by *eLife*. Your article has been reviewed by three peer reviewers, including Mani Ramaswami as the Reviewing Editor and Reviewer #1, and the evaluation has been overseen by a Reviewing Editor and Ronald Calabrese as the Senior Editor. The following individuals involved in review of your submission have agreed to reveal their identity: Baranidharan Raman (Reviewer #3).

The reviewers have discussed the reviews with one another and the Reviewing Editor has drafted this decision to help you prepare a revised submission.

Summary:

This study investigates how the contralaterally-projecting serotonin-immunoreactive deuterocerebral neuron (CSDn) functions in olfactory processing across two brain areas: the antennal lobe (AL), the first olfactory neuropil, and the downstream lateral horn (LH), which is involved in innate olfactory behavior. By employing 2-photon calcium imaging the author show that the CSDn processes in the antennal lobe are hyperpolarized during most odorant exposure, whereas the CSDn processes in the downstream lateral horn are depolarized during the same epochs. Interestingly, the AL responses seem to be uniform and independent of the odor applied, while the LH responses show some degree of odor-specificity. By correlating the odor-evoked activities of the CSDn in the LH to responses of PNs, the authors demonstrate that these patterns are highly correlated and that PNs most likely provide the synaptic input onto the CSDn in the LH. GRASP experiments between both neuronal populations yield a positive signal and provide further support for a direct synaptic connection between certain PNs and the CSDn. Moreover, by using a dataset derived from whole brain EM reconstructions, the authors show that the CSDn receives input from about 17 glomeruli from the AL.

Using an experimentally-constrained, multi-compartmental model that captures the passive properties of the serotonergic neuron to analyze whether the neurites in the AL and LH function as independent compartments within the CSDn, the authors show that transmission of electrical signals in this neuron is asymmetric. While hyperpolarization from the antennal lobe spreads to the somatic and LH segments, propagation of electrical signals from LH compartment to the AL/somatic segments is impeded. Using laser ablation to isolate CSDn processes in LH, they test a prediction from the model. Odor-evoked CSDn responses in LH are enhanced following severing of connection between the AL and LH (as negative contributions from the hyperpolarized processes in the AL are removed).

In summary, this study demonstrates that the CSDn shows distinct odor response properties in different brain regions. Odor-independent inhibition in the AL is likely mediated by multiglomerular inhibitory local interneurons and odor-specific excitation in the LH mediated by projection neurons. It also shows that inhibition from the AL propagates to probably suppress olfactory responses at the LH level.

Showing how different olfactory circuits may locally and differentially engage or disengage serotonergic modulation of stimulus-evoked responses, the findings of this study are very interesting and constitute an important advance on the authors previous paper in *eLife*. However, it could be improved and enhanced in the following ways.

Essential revisions:

1) It would be valuable to identify the synaptic partners of the CSDn in the LH to elucidate which responses are being suppressed at the LH level. The authors could use their EM reconstruction dataset to follow-up the neuronal circuitry within (and beyond) the LH.

2) It would also be interesting to know whether PNs and the CSDn have any synaptic contacts at the AL level and if so, what that means for the CSDn responses in the LH. These data may already be available from the GRASP experiments.

---

## [Author Response]

Essential revisions:1) It would be valuable to identify the synaptic partners of the CSDn in the LH to elucidate which responses are being suppressed at the LH level. The authors could use their EM reconstruction dataset to follow-up the neuronal circuitry within (and beyond) the LH.

This is a fantastic suggestion and is a significant component of the thesis work of one of the authors on this manuscript. The scope of that manuscript and analysis is extensive and includes a detailed description of the connectivity of the CSDn in the LH. That work will be submitted for publication within the year.

2) It would also be interesting to know whether PNs and the CSDn have any synaptic contacts at the AL level and if so, what that means for the CSDn responses in the LH. These data may already be available from the GRASP experiments.

Yes, this data was previously published in Coates et al., 2017 using the GRASP approach. That work revealed that CSDns receive input from very few ORNs and only the DM5 PN (confirmed with physiology). Consistent with our current stu et al., 1998dy, Coates et al. revealed that the predominant input to the CSDns was inhibitory via GABAergic interneurons. DM5 PNs can be reliably activated with apple cider vinegar, however this odor inhibits the CSDn. This is likely because apple cider vinegar also recruits sufficient global inhibition in the antennal lobe. To activate the CSDn, an odor must drive excitatory inputs into the neuron and recruit little global inhibition. Ammonia appears unique in this case as ammonia-sensitive ORNs synapse directly onto the CSDn and the odorant recruits little global inhibition. We now make a direct reference to the findings of Coates et al., 2017 in the Results section when describing our GRASP findings in the LH.